# Addressing the Complex Links between Psychopathy and Childhood Maltreatment, Emotion Regulation, and Aggression—A Network Analysis in Adults

**DOI:** 10.3390/bs14020115

**Published:** 2024-02-04

**Authors:** Diana Moreira, Candy Silva, Patrícia Moreira, Tiago Miguel Pinto, Raquel Costa, Diogo Lamela, Inês Jongenelen, Rita Pasion

**Affiliations:** 1Centro Regional de Braga, Universidade Católica Portuguesa, 4710-362 Braga, Portugal; candysilva348@gmail.com (C.S.); patriciaigmoreira@gmail.com (P.M.); 2Laboratory of Neuropsychophysiology, Faculty of Psychology and Educational Sciences, University of Porto, 4200-135 Porto, Portugal; 3Centro de Solidariedade de Braga/Projeto Homem, 4700-024 Braga, Portugal; 4Institute of Psychology and Neuropsychology of Porto—IPNP Health, 4100-341 Porto, Portugal; 5Observatory Permanent Violence and Crime (OPVC), FP-I3ID, Fernando Pessoa University, 4249-004 Porto, Portugal; 6HEI-Lab—Digital Human-Environment Interaction Labs, Lusófona University, 4000-098 Lisbon, Portugal; tiago.pinto@ulusofona.pt (T.M.P.); raquel.costa@ulp.pt (R.C.); diogo.lamela@ulusofona.pt (D.L.); ines.jongenelen@ulusofona.pt (I.J.); rita.pasion@ulusofona.pt (R.P.)

**Keywords:** childhood maltreatment, emotion regulation, psychopathy, aggression, nomological network

## Abstract

Childhood maltreatment is the strongest predictor of psychopathology and personality disorders across the lifespan and is strongly associated with a variety of psychological problems, namely, mood and anxiety disorders, behavioral and personality disorders, substance abuse, aggression, and self-harm. In this study, we aim to provide a comprehensive picture of the interplay between different traits of psychopathy and distinct dimensions of childhood maltreatment, emotion regulation, and aggression. Using a cross-sectional design, we employed correlational network analysis to explore the nomological network of psychopathy and provide a sample-based estimate of the population parameters reflecting the direction, strength, and patterns of relationships between variables. The sample consisted of 846 adults (71% females) who completed questionnaires measuring psychopathy, childhood maltreatment, emotion regulation, and aggression. The results highlight that disinhibition traits of psychopathy are the closest attributes of early experiences of abuse (but not neglect) in childhood and correlate with all dimensions of emotion regulation difficulties, being specifically associated with reactive aggression. Neglect was a unique attribute in the nomological network of meanness, with widespread correlations with emotion regulation difficulties but also an increased ability to engage in goal-directed behavior. Physical abuse was the only dimension of childhood adversity that was found to be intercorrelated with boldness and increased emotional regulation was found in this psychopathic trait. No significant associations were found between boldness, meanness, and aggression once shared variance with disinhibition was controlled. These results are discussed in terms of their implication for research and clinical practice.

## 1. Introduction

According to the World Health Organization, 300 million children regularly suffer physical punishment and/or psychological violence at the hands of parents and caregivers [1]. Childhood maltreatment is the most robust predictor of psychopathology and personality disorders across the lifespan [2], being closely associated with a wide spectrum of psychological problems, namely, mood and anxiety disorders, substance abuse, aggression, and self-harm [3,4,5,6].

Emotion regulation difficulties are one of the mechanisms that may account for this link [7,8,9,10]. Abuse and/or neglect by caregivers can negatively affect the development of socioemotional skills and regulatory capacity in children [9,11] since caregivers are not available to assist their children in modulating physiological arousal. By contrast, caregivers can exponentiate children’s negative emotionality [12]. Simultaneously, maltreated children are likely to allude to how their caregivers regulate their emotions, being exposed to ineffective strategies [9,10]. From another perspective, research shows that maltreatment can significantly impact individuals’ perceptions of the external world and their subsequent behavioral responses [13]. Children exposed to trauma show emotional hyperreactivity [14] such that salient negative cues in the environment (such as angry or fearful faces) elicit strong emotional responses [15,16]. Emotional hyperactivation to threat cues is adaptive when living in environments with high levels of danger as they allow individuals to anticipate threats and drive safety behaviors [7]. However, emotional hyperactivation can be maladaptive in environments where danger is minimal, increasing the chances of implementing defensive–aggressive strategies since the intensity and durability of this emotional response can be disproportional and more difficult to regulate [8].

In sum, previous findings highlight the interplay between childhood maltreatment and emotion dysregulation, incorporating emotional reactivity to distress as an operating vulnerability factor for psychopathology and personality disorders [7,8,9,10]. However, under the scope of the current work, this association can and shall be debated in psychopathy to provide an in-depth understanding of its causal chains, especially because studies focusing on this personality disorder report reduced (not increased) sensitivity to threat cues, making it tempting to preclude that individuals with more psychopathic traits are less likely to display negative affectivity, emotional regulation difficulties, and anger outbursts [17,18,19,20,21]. This directly conflicts with the existing literature showing a positive association between childhood abuse, psychopathy, difficulties in emotion regulation, and aggression [22,23,24,25,26,27,28,29,30], revealing the need to study how these variables may show differential associations when different phenotypic expressions of psychopathy are considered.

The Triarchic Model of Psychopathy [31,32] provides a nuanced perspective to explain how psychopathy can be related to both difficulties and preserved emotional regulation, especially when there is a history of previous abuse and neglect. This model defines three distinct phenotypic expressions of psychopathy: (1) disinhibition, which is a nexus of impulsivity and negative affectivity; (2) meanness, which represents agentic disaffiliation and empathic deficits; and (3) boldness, which is characterized by low-stress reactivity and is expected to be a more adaptive expression of psychopathy. The different configurations of the proposed phenotypes make it possible to accommodate the idea that distinct psychopathic profiles may exist, and, therefore, that they can be related to fundamental differences in developmental experiences, emotional regulation, and aggression.

Developmental factors, namely, adversity early in life, may indeed contribute to explaining the different phenotypic configurations within the psychopathic personality structure and aggression-related outcomes. For instance, meanness and disinhibition explain antisocial manifestations in psychopathy and are thought to share the same etiological pathway. Disinhibition-related traits are the closest correlates of childhood maltreatment [30] with deficits in emotional regulation, low effortful control, and high automatic reactivity to negative cues [24,29,31,32]. For instance, aggression in disinhibition assumes predominantly reactive/impulsive forms to a perceived threat [22,23,24,25,26,27,28,33,34,35,36]. Meanness traits also seem to be an outcome of processes of socialization that have failed, namely, a failure to develop secure attachments based on emotional caring [29,31,32,37]. Meanness encompasses features such as disdain for close attachments, shallow concern for the feelings of others, cruelty, exploitativeness, and premeditated/cold-blooded forms of aggression [22,24,25,27,28,33,34,35,36,38]. Boldness, in turn, is uncorrelated with disinhibition but shares the low fear etiological pathway with meanness [31,32,39]. Low fear has a genetic-based etiological pathway that reflects the failure to learn from experience and underlies low-stress reactivity [40,41,42,43,44,45,46,47,48]. As such, boldness entails characteristics like emotional resiliency, social assertiveness, lack of anxiety, tolerance of stressful events, and reduced threat sensitivity [31,39,49]. Being a more adaptive expression of psychopathy, it is proposed that developmental factors (e.g., secure attachment, high executive functioning and cognitive processing) may protect individuals with higher scores for boldness (and low disinhibition and moderate meanness scores) from engaging in disruptive–antisocial behaviors [31,32,36,38,50,51,52]. As such, while meanness and disinhibition phenotypic expressions related to externalizing and antisocial behaviors may be rooted in childhood adversity, optimal developmental environments for boldness can reverse or attenuate low-fear deficits, reducing the likelihood of aggression outcomes and setting the stage to develop more positive relationships with others in the future [31,32,39].

### Current Study

From the reviewed literature, one can assert that psychopathic trait dimensions exhibit differential associations with experiences of childhood abuse and neglect, emotion regulation, and aggression. Thus, addressing the complex relationships between all these variables can be a major contribution to the literature in this field. Indeed, research tends to explore psychopathy as a global score and associate its different phenotypic expressions with single outcomes. To the best of our knowledge, previous studies have not yet provided clear indications as to whether distinctive components of emotion dysregulation and aggression can be differentially related to psychopathic traits and the role of childhood adverse experiences in this complex pattern of associations.

In the current study, we aim to provide a clearer picture of the interplay between dimensions of childhood abuse and neglect (sexual, physical, and emotional abuse; physical and emotional neglect), different traits of psychopathy (disinhibition, boldness, and meanness), dimensions of emotion regulation (nonacceptance of emotional responses, difficulty engaging in goal-directed behavior, impulse control difficulties, limited access to emotion regulation strategies, lack of emotional awareness and clarity), and aggression outcomes (premeditated and reactive aggression). Specifically, we intend to (1) calculate correlations between each dimension/subscale entered in the model and (2) assess partial correlations that emerge from these individual links when all other variables in the model are taken into account. For instance, we will try to address these complex links by using correlational network analysis, which offers a fine-grained approach to analyzing complex relationships between variables, uncovering intricate patterns of relationships between them. In network modeling, all variables are examined simultaneously within a single model without specifying a priori any pattern of associations, which allows for a comprehensive understanding of their interdependencies. Moreover, it provides insights into the relative importance of each variable within the network, which is of critical variance to target variables that require further investigation and may be useful for intervention.

Exploring the nomological network of the Triarchic Model of Psychopathy [31] regarding childhood abuse, emotion regulation, and aggression dimensions, from the available literature, it is possible to hypothesize the following: (H1) Disinhibition is the strongest correlate of childhood maltreatment and emotion regulation difficulties and is linked to reactive aggression. (H2) Meanness correlates with childhood maltreatment, emotion regulation difficulties, and premeditated aggression. (H3) Boldness is not associated with childhood maltreatment, emotion regulation difficulties, or aggression dimensions.

## 2. Methods

### 2.1. Study Design

Our study used quantitative, survey-based research in an attempt to provide an accurate estimate of the relationships between the variables of interest: psychopathy, childhood maltreatment, emotional regulation, and aggression.

Specifically, its cross-sectional correlational design allowed us to examine multiple constructs and their relationships in a specific timeframe [53]. Owing to the fact that network analysis visually represents the pattern of relationships between variables, we chose this statistical procedure to quantify it and provide a sample-based estimate of the population parameters reflecting the direction, strength, and patterns of these relationships [54].

### 2.2. Participants

We recruited the sample from the general population based on minimal inclusion criteria (i.e., fluent in Portuguese and aged equal to or older than 18 years). Following recent assumptions on the dimensional structure of psychological and psychopathological phenomena, we recruited the sample from the community and defined minimal exclusion criteria to increase the cohort approximation due to its complex nature and avoid interpretative conundrums in terms of its dimensional and continuous character [55,56].

The sample (see Table 1) consisted of 846 participants aged between 18 and 87 years (*M*_age_ = 30.9, *SD* = 0.49). Most participants were Portuguese (98.81) and had secondary or high education (85.46%). More than half were female (70.69%) and single (63.83%). Regarding exposure to early aversive experiences, 32.4% of adults reported having experienced emotional abuse, 19.3% reported experiencing physical abuse, 13.7% reported experiencing sexual abuse, 35.1% reported experiencing emotional neglect, and 18.1% reported experiencing physical neglect in childhood.

### 2.3. Measures 

#### 2.3.1. Psychopathy

The Triarchic Psychopathy Measure (TriPM; Patrick, unpublished, Portuguese version by [36]) was used to assess psychopathy. It is a self-reported instrument consisting of 58 items that classifies the three phenotypic constituents of psychopathy in adults: boldness (α = 0.63, e.g., the ability to remain calm in threatening situations), meanness (α = 0.85, e.g., emotional detachment), and disinhibition (α = 0.84, e.g., poor regulation of negative affect). These items are rated on a 4-point Likert-type scale ranging from 3 (true) to 0 (false).

#### 2.3.2. Childhood Abuse and Neglect

The Adversity History in Childhood Questionnaire (ACE; [57], Portuguese version by [58]) and the Childhood Trauma Questionnaire (CTQ; [59]) were used to assess childhood abuse and neglect. The ACE is a self-report questionnaire for the adult population that assesses the occurrence of adversity experiences in childhood: emotional abuse (α = 0.76, e.g., insults and humiliation), physical abuse (α = 0.79, e.g., throwing things and hitting), sexual abuse (α = 0.80, e.g., sexualized touches), emotional neglect (α = 0.78; e.g., lack of emotional care), and physical neglect (α = 0.54, e.g., lack of medical care). The CTQ (CTQ; [59]) provides a reliable measurement of the same type of abuse evaluated in ACE: emotional (α = 0.69), physical (α = 0.68), and sexual abuse (α = 0.83) and emotional (α = 0.79) and physical neglect (α = 0.64). 

Both measures are rated on a 5-point Likert scale (never/never true to very often/very often true). It was understood that the experience of adversity had occurred if the person answered affirmatively to at least one question of the dimension being analyzed.

#### 2.3.3. Emotion Regulation

The Difficulties in Emotional Regulation Scale (DERS; [60], Portuguese version by [61]) was used to assess emotion regulation. It classifies the typical levels of emotional dysregulation in individuals. It comprises 36 items scored on a 5-point Likert-type scale ranging from 1 (almost never) to 5 (almost always) and evaluates six domains: non-acceptance of negative emotions (α = 0.90, e.g., “When I’m upset, I feel ashamed with myself”), the inability to engage in goal-directed behaviors when experiencing negative emotions (α = 0.73, e.g., “When I’m upset, I have difficulty concentrating”), difficulties in controlling impulsive behaviors when distressed (α = 0.81, e.g., “When I’m upset, I feel out of control”), limited access to emotion regulation strategies that are perceived as effective (α = 0.87, e.g., “When I’m upset, I’ll remain that way for a long time”), lack of emotional awareness (α = 0.81, e.g., “I pay attention to what I feel”), and lack of clarity (α = 0.60, e.g., “I have no idea of what I’m thinking”).

#### 2.3.4. Aggression

The Impulsive and Premeditated Aggression Scale (IPAS; [62], Portuguese version by [63]) was used to assess aggression. It is a self-rating scale that classifies impulsive (α = 0.91, e.g., “Anything could have set me off prior to the accidents”) and premeditated (α = 0.87, e.g., “I feel my actions were necessary to get what I wanted”) forms of aggression. The IPAS comprises 30 items rated on a 5-point Likert-type scale (1 = strongly disagree to 5 = strongly agree).

### 2.4. Procedures

Data collection in an online survey was performed through digital social networks and snowballing procedures. Participants were required to provide their informed consent, without which they could not proceed to complete the protocol. Participants were also informed about the scope and objectives of the study, as well as the confidentiality and anonymity of the data collected. No financial compensation was awarded for participation in this study.

This study was approved by the Scientific Committee of the Universidade Católica Portuguesa, Faculty of Philosophy and Social Sciences, Centre for Philosophical and Humanistic Studies, which is responsible for ethical verification.

### 2.5. Network Analysis

The LASSO Gaussian graphical model [64,65] was used to explore intercorrelations between childhood maltreatment (emotional abuse, physical abuse, sexual abuse, emotional neglect, and physical neglect), emotion regulation (nonacceptance of emotional responses, lack of emotional awareness, impulse control difficulties, difficulties engaging in goal-directed behavior, lack of emotional clarity, and limited access to emotion regulation strategies), psychopathy traits (meanness, boldness, and disinhibition), and aggression (impulsive and premeditated aggression). As such, the current network was composed of nodes (i.e., each subscale included in the model) and edges (i.e., connections among subscales included in the model). Associations were undirected, but it is important to acknowledge that childhood trauma temporally preceded all other nodes. LASSO is a powerful tool for fast high-dimensional networks [65]. It penalizes the sum of absolute parameter values such that they will often equal 0, that is, the resulting model is almost sparse (i.e., only a few parameters are around nonzero) but prevents model overfitting. Most importantly, if the true network structure is sparse, LASSO will return fewer false positives. In the current model, all variables (i.e., subscales) were z-scored, considering the questionnaires they were part of and that there were no missing values in individual responses.

We used the Extended Bayesian Information Criterion (EBIC) to compute partial correlation coefficients. EBIC returns a parsimonious network model since the association between two nodes is controlled for the influence of all other variables [65]. The thickness of an edge graphically represents the magnitude of this association. Missing edges indicate the independence of two nodes after conditioning on all other nodes. The EBIC parameter to set the degree of regularization applied to sparse correlations was set to γ = 0.5 [65,66]. To further explore how well a node was directly connected to other nodes in the network structure, we explored strength, betweenness, and closeness [67]. Strength identifies the node with the highest degree of association with other nodes in the network while considering the edge weights. Betweenness quantifies the relative number of shortest paths passing through a specific node, showing how a specific node can influence the information flow between nondirectly connected nodes. Closeness detects what node information can reach other nodes quickly. To evaluate the accuracy of edge estimates, nonparametric bootstrapping (1000 samples) was used. All analyses were performed in JASP 0.17.2.1 (bootnet and qgraph package in R) using the EBICglasso estimator [65,68].

## 3. Results

The network (see Figure 1) and weights matrix (see Table 2) are presented below. The network was constituted of 64 out of 120 nonzero edges, with a sparsity of 0.47 (i.e., 66 edges were not selected by LASSO). The different subscales tended to cluster together, organized around the main scale. Betweenness and closeness centrality measures showed high stability. Both displayed higher centrality scores in the difficulties in emotion regulation subscales of impulse control difficulties, emotional clarity, and emotion regulation strategies. Regarding strength, the highest scores were found in difficulties in emotion regulation strategies, emotional abuse, and meanness. Across the three centrality measures, we can observe that the (difficulties in) emotion regulation subscale is a dimension that systematically emerged in the model. Model estimation was fairly accurate, with a noticeable proportion of edges where 95% CIs did not overlap.

### 3.1. Nomological Network

#### 3.1.1. Disinhibition

Emotional (*r_partial_* = 0.061), physical (*r_partial_* = 0.022), and sexual abuse (*r_partial_* = 0.089) exhibited intercorrelations with sexual abuse, demonstrating the highest degree of association. Associations between disinhibition and neglect, either emotional or physical, were at the zero-order. Regarding emotion regulation difficulties, disinhibition traits related positively with all dimensions, but stronger correlations were found with impulse control difficulties (*r_partial_* = 0.118) and lack of emotional awareness (*r_partial_* = 0.085). Disinhibition was uniquely associated with reactive aggression (*r_partial_* = 0.003).

#### 3.1.2. Meanness

Sexual abuse (*r_partial_* = 0.065) and physical neglect (*r_partial_* = 0.062) were correlated with meanness traits. No other associations were found between meanness and childhood adversity subscales. Meanness was associated with lack of emotional clarity (*r_partial_* = 0.120) (also impulse control difficulties, *r_partial_* = 0.118, and lack of emotional awareness, *r_partial_* = 0.085); however, a negative association was found with goal-directed behavior (i.e., great ability to engage in goal-directed behavior, *r_partial_* = −0.062). No associations were found with aggression.

#### 3.1.3. Boldness

Physical abuse was the only dimension of childhood adversity that was found to be intercorrelated with boldness traits (*r_partial_* = 0.025). Regarding subscales of emotion regulation, only negative associations were observed. The strongest associations were found between boldness and more access to emotion regulation strategies (*r_partial_* = −0.146) and higher emotional awareness (*r_partial_* = −0.064). No significant associations were reported relative to aggression outcomes.

## 4. Discussion

In the current study, we aimed to explore the nomological network of psychopathy by providing a comprehensive picture of the interplay between dimensions of childhood maltreatment, psychopathic traits, emotional regulation, and aggression outcomes. Using network modeling, we were able to isolate specific associations with psychopathic traits once the effects of other variables were accounted for. The results are described in detail below, considering each of the hypotheses being tested.

### 4.1. Disinhibition

Regarding the first hypothesis (disinhibition is the strongest correlate of childhood abuse and difficulties in regulating emotions and is linked to reactive aggression), we can say that it was verified in this study, excluding the variable neglect, since it was not significant.

In line with previous research [30,69], disinhibition traits of psychopathy emerged as the closest attributes of early experiences of abuse in childhood. They correlated with all forms of abuse (sexual, physical, and emotional), although they showed no significant relationships with neglect experiences. This result does not suggest that individuals characterized by high disinhibition–impulsive traits do not experience neglect in childhood, namely because several forms of abuse and neglect are expected to co-occur [70,71]. What our results seem to suggest is that when controlling for other phenotypic expressions of psychopathy (e.g., meanness), disinhibition seems to correlate with more extreme forms of childhood maltreatment [30]. In other words, adults who show the greatest tendencies toward impulsive behavior and hostility are also more likely to report experiencing maltreatment as a child and, consequently, experience less parental warmth, less maternal protection, and greater feelings of rejection by parents [69].

Evidence from emotion regulation strategies and patterns of aggression further contribute to understanding the impact of sexual, physical, and emotional abuse on adults reporting high disinhibition. In our study, disinhibition traits covaried with all dimensions of emotion regulation difficulties, and especially with impulse control deficits and a lack of emotional awareness. The convergent link between disinhibition and impulse control difficulties was expected due to the definition of the constructs themselves (i.e., impulse control difficulties are an inherent characteristic of impulsive and disinhibited behavior). Nonetheless, other emotional regulation deficits, such as a lack of emotional awareness, add new dimensions to the nomological network of disinhibition.

Overall, emotion regulation difficulties can help to explain the associations between disinhibition and reactive aggression found in the current study. Given that experiences of child maltreatment are characterized by high negative affectivity and compromise the development of socioemotional skills in children [9,11], it remains possible that individuals reporting high disinhibition are likely to exhibit a lesser ability to regulate negative emotions and stress reactions while displaying high automatic reactivity to negative cues [24,31,32]. From a developmental perspective, heightened reactivity to internal negative affect and external environmental cues, when coupled with deficits in emotion regulation and inhibitory control, is likely to influence the interpretation of social information and, consequently, behavioral responses. For instance, the lack of positive interactions with caregivers can reduce the chances of children effectively managing emotional stimulation, possibly increasing externalizing reactions and behaviors [7,9,10]. At the same time, an increased ability to identify hostile cues to which they have been repeatedly exposed and that represent (negative) salience [72,73] can lead individuals to be more prone to perceiving future situations as hostile and respond accordingly—even if stimuli are ambiguous [74,75]. Hostile attributions are actually described as a nuclear component of cognitive models exploring the development of aggressive behavior and are thought to have a link with previous exposure to violence/abuse [72,76,77].

In light of our study, aggression and disinhibition indeed assume reactive/impulsive forms to a perceived threat and an inability to regulate anger outbursts and intense negative affect [22,23,24,25,26,27,28,33,34,35,36]. The distinction between reactive and premeditated aggression is an important one because reactive violence in disinhibition is expected to occur in response to provocation (or stimuli perceived as hostile) and is marked by an unplanned response that the individual finds difficult to suppress given the experience of high negative affect [22,24,27,28,31,32,78].

### 4.2. Meanness

Relative to the second hypothesis (meanness correlates with childhood maltreatment, emotion regulation difficulties, and premeditated aggression), it was partially confirmed.

Supporting the assumption that meanness and disinhibition are the main phenotypic expressions of psychopathy rooted in childhood adversity, meanness correlated positively with sexual abuse and physical neglect. Sexual abuse seems to be a shared experience with disinhibition, but (physical) neglect was a unique attribute in the nomological network of meanness. This is an important aspect to deepen our understanding of early experiences of neglect, personality traits characterized by a lack of empathy, and emotion regulation, as reported in the current study.

Emotional neglect refers to a pattern of failing to meet children’s emotional needs, limiting their access to experiences of positive affect [59]. While disinhibition is associated with more experiences of abuse and exposure to (intense negative) affect, individuals reporting high meanness may have fewer chances to have emotional experiences overall. This supports some different configurations between disinhibition and meanness that are assumed in their theoretical conceptualizations [31,32]. Individuals who are emotionally neglected, like in meanness, may struggle to express their feelings appropriately, which can increase the likelihood of having pent-up anger and internalizing frustration, whereas, in disinhibition, the pattern seems to be toward externalization, given the greater exposure to experiences of abuse.

Considering the impact of maltreatment experiences on emotional regulation, as observed with disinhibition, one would expect meanness to correlate with emotional regulation difficulties. This was true for lack of emotional clarity and awareness and lack of inhibitory control (similarly to disinhibition), but an opposite pattern was found in goal-directed behavior. As such, individuals reporting high meanness seem to display a strong ability to engage in goal-directed behavior. When combined with lack of empathy and cruelty traits, this goal-directed behavior will probably progress toward seeking control and power over others, showing manipulative tendencies as a means of asserting dominance, or inflicting harm, as described in the theoretical conceptualization of meanness [31,32].

However, it should be noted that meanness did not correlate with premeditated aggression, which contrasts with previous research [22,24,27,28,31,32,78]. Since our model comprised a larger variety of variables than previous studies, we need to acknowledge that emotion regulation strategies can have a more widespread association with aggression outcomes than with individual differences in personality per se.

### 4.3. Boldness

The last hypothesis stated that boldness was associated with childhood maltreatment, emotion regulation difficulties, and aggression dimensions. This was verified, except for physical abuse.

Previous studies found that boldness was negatively related to emotional abuse and physical neglect, suggesting that individuals who reported high adaptive traits of psychopathy such as a higher resilience to stress, low anxiety, and less ability to step away from their comfort zone are also less likely to report experiences of childhood maltreatment [69]. In the current study, physical abuse was the only dimension that related to boldness. Although a positive correlation emerged, our results seem to suggest that this phenotypic expression of psychopathy yielded the weakest associations with childhood maltreatment when compared with disinhibition and meanness. Still, physical abuse is part of the nomological network of boldness and may have a negative impact on human development.

Overall, we gathered evidence that factors other than early developmental experiences may be implicated in the adaptive adjustment that is expected to be found in individuals with high boldness throughout the lifespan. High executive functioning and efficient cognitive processing have been presented before as important factors that may prevent individuals displaying high boldness traits from engaging in disruptive–antisocial behaviors [31,32,36,38,50,51,52].

The current study supports this evidence and highlights that emotional regulation can play an important role in buffering the detrimental effects of physical abuse. Boldness showed a negative association with emotion regulation difficulties, that is, this phenotype seems to be mainly related to increased access to and abilities to use different strategies to regulate emotions (and behavior), especially because the strongest patterns of correlations were found in strategies and awareness dimensions. This may, in turn, explain the absence of significant relationships with aggression outcomes.

## 5. Conclusions and Practical Implications

Despite the relevant contributions to the existing literature, our study has limitations that should be acknowledged. First, the sample size was relatively small compromising, for example, the analysis with gender as a moderator. On this topic, the sample size was not calculated to be representative of the population and was recruited by convenience. Considering the characteristics of the included participants, we do not expect our results to be generalizable to other contexts (e.g., clinical and forensic samples) and may also be limited regarding understanding the nomological network of psychopathy in males. The cross-sectional, correlational design we used further prevents us from inferring any nexus of causality. For instance, lower reliability coefficients in several subscales of childhood maltreatment (and especially in physical neglect) may underscore that the scale does not collect the necessary amount of variability in item responding (i.e., from extreme to lower values in childhood maltreatment experiences. Finally, there is a call on the need to better conceptualize and operationalize childhood maltreatment experiences by emphasizing the importance of considering the perspectives of children and survivors [79]. Related to this, we also measured childhood maltreatment in a retrospective way, which yielded several limitations (e.g., recall biases, difficulty in assessing severity, and misclassification of different maltreatment experiences). Future research should address these limitations.

Overall, the results show that the nomological network is unique to each phenotypic expression and provides important insights for personalized interventions. Through the network analysis, we were able to perceive the connections between variables, see their strength, and learn from that by comprehending the intricate dynamics involving child maltreatment, emotional regulation, and aggression, all associated with each unique phenotypic expression, which may inform tailored psychosocial interventions that can more effectively address the clinical challenges presented by individuals exhibiting psychopathic traits. Due to the heterogeneity of the psychopathic personality structure, related to the distinct configuration of its phenotypes, professionals working with this population may need to develop different intervention strategies, each one tailored to the specific profile (e.g., targets and intensity). For example, for individuals scoring high in disinhibition (and low-to-moderate in the other dimensions of psychopathy), as measured by a psychological assessment protocol, mental health professionals might opt to explore developmental experiences in more depth and develop strategies to ameliorate emotional regulation difficulties in case they interfere with the adjustment of the individual under intervention.

## Figures and Tables

**Figure 1 behavsci-14-00115-f001:**
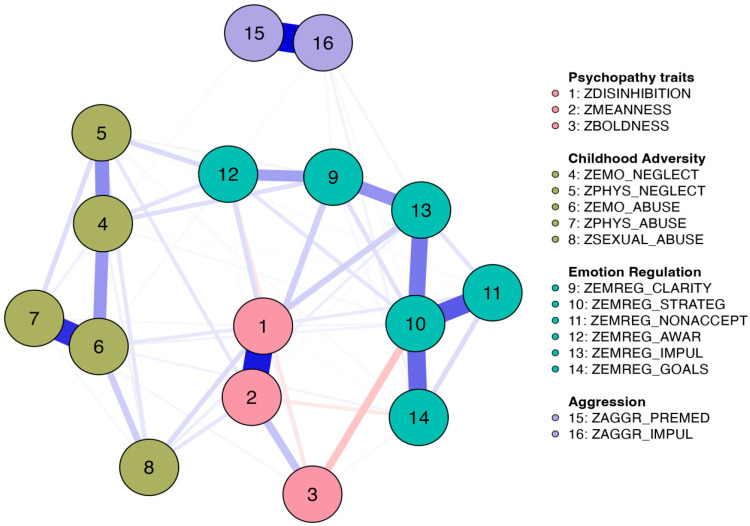
Network model. *Note.* The higher the thickness of an edge, the higher the magnitude of the association conditioning on all other nodes. Missing edges indicate the independence of two nodes after conditioning on all other nodes. Blue edges represent positive associations between nodes. Red edges represent negative associations between nodes. All nodes were converted to z scores. Red nodes represent psychopathic phenotypic expressions: disinhibition (1), meanness (2), and boldness (3). Brown edges illustrate maltreatment experiences, as measured retrospectively in childhood: emotional neglect (4), physical neglect (5), emotional abuse (6), physical abuse (7), and sexual abuse (8). Green nodes represent emotional regulation subscales: lack of emotional clarity (9), limited access to emotion regulation strategies (10), non-acceptance of negative emotions (11), lack of emotional awareness (12), difficulties in controlling impulsive behaviors (13), and inability to engage in goal-directed behaviors (14). Purple edges show aggression outcomes: premeditated (15) and impulsive (16) aggression.

**Table 1 behavsci-14-00115-t001:** Descriptives (*N* = 846).

Variable	Group	*n*	*%*
Age	18–31	555	65.60
		138	
		111	
	32–45	35	16.31
	46–59	7	13.12
	60–73	598	4.14
	74–87	248	0.83
Gender	Woman	836	70.69
	Man	6	29.31
Nationality	Portuguese	1	98.81
	Other (e.g., Brazilian)	1	0.72
	Missing	1	0.47
Marital Status	Single	1	63.83
	Married	4	23.41
	Cohabiting	540	6.38
	Divorced	198	3.90
	Widower	54	0.71
	Missing	33	1.77
Education	Did not finish primary school	6	0.12
	1st to 4th grade	15	4.73
	5th to 6th grade	1	2.60
	7th to 9th grade	40	7.09
	Higher school degree	22	44.44
	Honours degree	60	27.07
	Master’s degree	376	9.46
	Doctorate degree	229	4.49
		80	
		38	

**Table 2 behavsci-14-00115-t002:** Weights matrix (partial correlations) of the network model.

	Network															
Variable	Disinhibition	Meanness	Boldness	Emotional Neglect	Physical Neglect	Emotional Abuse	Physical Abuse	Sexual Abuse	Clarity	Strategies	Nonacceptance	Awareness	Impulse Control	Goals	Premeditated Aggression	Impulsive Aggression
Disinhibition	0.000	0.567	0.029	0.000	0.000	0.061	0.022	0.089	0.002	0.025	0.055	0.085	0.118	0.008	0.000	0.003
Meanness	0.567	0.000	0.150	0.000	0.062	0.000	0.000	0.065	0.120	0.000	0.000	0.009	0.014	−0.062	0.000	0.000
Boldness	0.029	0.150	0.000	0.000	0.000	0.000	0.025	0.000	−0.005	−0.146	0.000	−0.064	0.000	−0.021	0.000	0.000
Emotional neglect	0.000	0.000	0.000	0.000	0.290	0.262	0.036	0.034	0.082	0.019	−0.009	0.073	0.000	0.000	0.000	0.000
Physical neglect	0.000	0.062	0.000	0.290	0.000	0.045	0.085	0.065	0.029	0.000	−0.015	0.094	0.000	−0.020	0.000	0.000
Emotional abuse	0.061	0.000	0.000	0.262	0.045	0.000	0.516	0.126	0.000	0.032	0.000	−0.017	0.000	0.030	0.000	0.012
Physical abuse	0.022	0.000	0.025	0.036	0.085	0.516	0.000	0.021	0.000	0.000	0.000	0.000	0.000	0.000	0.014	0.000
Sexual abuse	0.089	0.065	0.000	0.034	0.065	0.126	0.021	0.000	0.000	0.000	0.000	0.000	0.045	0.000	0.000	0.000
Clarity	0.002	0.120	−0.005	0.082	0.029	0.000	0.000	0.000	0.000	0.078	0.000	0.237	0.275	0.023	0.000	0.018
Strategies	0.025	0.000	−0.146	0.019	0.000	0.032	0.000	0.000	0.078	0.000	0.417	0.067	0.341	0.380	0.000	0.015
Nonacceptance	0.055	0.000	0.000	−0.009	−0.015	0.000	0.000	0.000	0.000	0.417	0.000	0.000	0.073	0.091	0.000	0.000
Awareness	0.085	0.009	−0.064	0.073	0.094	−0.017	0.000	0.000	0.237	0.067	0.000	0.000	0.000	0.000	0.000	0.000
Impulse control	0.118	0.014	0.000	0.000	0.000	0.000	0.000	0.045	0.275	0.341	0.073	0.000	0.000	0.085	0.000	0.029
Goals	0.008	−0.062	−0.021	0.000	−0.020	0.030	0.000	0.000	0.023	0.380	0.091	0.000	0.085	0.000	0.000	0.034
Premeditated aggression	0.000	0.000	0.000	0.000	0.000	0.000	0.014	0.000	0.000	0.000	0.000	0.000	0.000	0.000	0.000	0.638
Impulsive aggression	0.003	0.000	0.000	0.000	0.000	0.012	0.000	0.000	0.018	0.015	0.000	0.000	0.029	0.034	0.638	0.000

*Note.* Negative partial correlations in emotion regulation subscales should be interpreted as less difficulty in emotion regulation.

## Data Availability

Dataset available on request from the authors.

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
