# Peer review of "Addressing the Complex Links between Psychopathy and Childhood Maltreatment, Emotion Regulation, and Aggression—A Network Analysis in Adults"

_behavsci, 2024, doi:10.3390/bs14020115_

Round 1

Reviewer 1 Report

Comments and Suggestions for Authors

This manuscript employs Network Analysis to explore the relationship between several variables, including Childhood Maltreatment, Emotion Regulation, Aggression, and different dimensions of Psychopathy. Generally, the paper is well-written and demonstrates an innovative perspective. There are several minor points to discuss with the authors.

Throughout, Psychopathy appears to be a key outcome variable around which all other variables are organized. Please rephrase your Title to highlight this.

Physical neglect (pp.5, line 210) has a lower reliability coefficient. Why?

The practical implications of this study need to be expanded.

Author Response

Review

Comment

This manuscript employs Network Analysis to explore the relationship between several variables, including Childhood Maltreatment, Emotion Regulation, Aggression, and different dimensions of Psychopathy. Generally, the paper is well-written and demonstrates an innovative perspective. There are several minor points to discuss with the authors.

Throughout, Psychopathy appears to be a key outcome variable around which all other variables are organized. Please rephrase your Title to highlight this.

Thank you very much for your comment and for the time you spent reviewing our work. We followed your suggestions and corrected the title. Please see page 1.

Physical neglect (pp.5, line 210) has a lower reliability coefficient.

Why?

Although we found a smaller value for physical neglect, we would like to acknowledge that this alpha value falls into the range of others – at least in a qualitative way, showing some scales do not perform perfectly regarding their internal consistency. We think it is because our sample does not show extreme scores on childhood adversity and now included this as a limitation of the current study.

The practical implications of this study need to be expanded.

Thank you for your thorough review. Changes have been made in the conclusion section.

Reviewer 2 Report

Comments and Suggestions for Authors

Dear authors of the manuscript "Addressing the Complex Links Between Childhood Maltreatment, Emotion Regulation, Psychopathy, and Aggression - A Network Analysis",

I hope this letter finds you well. I want to begin by expressing my admiration for your research, which I consider an extremely interesting and currently relevant contribution. The way you approach the relationship between child maltreatment, emotional regulation, psychopathy, and aggression through network analysis is a valuable and promising perspective.

However, as a reader interested in your work, I would like to make some suggestions that, in my opinion, could improve the quality and clarity of the manuscript:

- Abstract: In the abstract, I suggest that you limit yourself to stating the objective of the research rather than including the hypotheses of the study. In addition, it would be beneficial to include the type or design of study in this section. The ideal structure of the abstract could be as follows: Introduction of the subject matter to be addressed, Objective of the study, Design of the study, Main results, and Conclusions.

- Methods: The section "2. Methods" seems to be somewhat disorganized. I suggest reorganizing this section into subsections that clearly describe the study design, participants, instruments used, procedure, and data analysis. This will facilitate understanding of the methodological approach of your research.

- Results: In the results section, it would be beneficial to provide relevant statistics to support the results presented. This would help readers evaluate the robustness of the findings and their statistical significance.

- Discussion: In the discussion section, instead of organizing it into "subsections," I suggest addressing each of the initial hypotheses sequentially, discussing whether or not they were met. This structure would allow for a clearer and more coherent presentation of the results in relation to the initial expectations of the research.

- Conclusions: In the conclusions section, it would be useful to highlight the limitations of the research, identify possible future lines of research, and highlight the specific contributions of your study to the scientific community. This will help contextualize the results and clarify their relevance to the field.

In summary, their research is valuable and promising, and these suggestions are presented with the goal of improving the presentation and comprehensibility of the manuscript. I appreciate your dedication to this important topic and hope to see a revised version of your work in the future.

Congratulations on your work, 

Best regards. 

Author Response

Review

Comment

I hope this letter finds you well. I want to begin by expressing my admiration for your research, which I consider an extremely interesting and currently relevant contribution. The way you approach the relationship between child maltreatment, emotional regulation, psychopathy, and aggression through network analysis is a valuable and promising perspective.

However, as a reader interested in your work, I would like to make some suggestions that, in my opinion, could improve the quality and clarity of the manuscript:

- Abstract: In the abstract, I suggest that you limit yourself to stating the objective of the research rather than including the hypotheses of the study. In addition, it would be beneficial to include the type or design of study in this section. The ideal structure of the abstract could be as follows: Introduction of the subject matter to be addressed, Objective of the study, Design of the study, Main results, and Conclusions.

Thank you very much for your comment and for the time you spent reviewing our work. We completely agree with your comment and reviewed substantially the abstract. Please see page 1.

- Methods: The section "2. Methods" seems to be somewhat disorganized. I suggest reorganizing this section into subsections that clearly describe the study design, participants, instruments used, procedure, and data analysis. This will facilitate understanding of the methodological approach of your research.

Thank you for your comment, changes have been made in this section.

- Results: In the results section, it would be beneficial to provide relevant statistics to support the results presented. This would

help readers evaluate the robustness of the findings and their statistical significance.

This information was provided. Please see the result section.

- Discussion: In the discussion section, instead of organizing it into "subsections," I suggest addressing each of the initial hypotheses sequentially, discussing whether or not they were met. This structure would allow for a clearer and more coherent presentation of the results in relation to the initial expectations of the research.

Thank you for your valuable comment. We have made some changes to meet your suggestions. Please see the discussion section.

- Conclusions: In the conclusions section, it would be useful to highlight the limitations of the research, identify possible future lines of research, and highlight the specific contributions of your study to the scientific community. This will help contextualize the results and clarify their relevance to the field.

In summary, their research is valuable and promising, and these suggestions are presented with the goal of improving the

presentation and comprehensibility of the manuscript. I appreciate your dedication to this important topic and hope to see a revised version of your work in the future.

Thank you for making us aware of this issue. All changes were made, also considering other reviewers’ suggestions. Please see “Conclusions” section.

Reviewer 3 Report

Comments and Suggestions for Authors

Authors should improve

1.  The title should have information about the characteristics of the sample (adults, people over 18 years old, adult women, etc.).

2.  The objective(s) of the study should be in the abstract.

3. It would be advisable to improve the presentation of the contents in the introduction in order to avoid repeating studies and authors in the discussion. All this to give clarity to both sections.

4. The way in which the sample is configured must have bibliographical references to give it validity. 

5. The authors must provide information on the representativeness of the sample.

6. The general and specific objectives of the study should be explained with greater precision (only mentioned in line 182, without specifying).

7. Table 2 is too broad. The results should be presented more precisely and described in relation to the data in the tables. 

8. The conclusions are too few. The proposals they mention should be expanded and more detailed with more information.

Author Response

Review

Comment

1. The title should have information about the characteristics of the sample (adults, people over 18 years old, adult women, etc.).

Thank you very much for your comment and for the time you spent reviewing our work. We completely agree with you and made some changes in the title.

2. The objective(s) of the study should be in the abstract.

Thank you for your comment. We agree and made the changes that you suggested, also following other reviewers’ commentaries.

3. It would be advisable to improve the presentation of the contents in the introduction in order to avoid repeating studies and authors in the discussion. All this to give clarity to both sections.

We reviewed the introduction and the discussion sections to avoid (or at least reduce) overlap in the main ideas. We found that a specific paragraph in the discussion was quite similar to an idea exposed in the introduction – and we rephrased it (highlighted in green). Thank you so much for your careful reading and for alerting us. However, we would like to repeat studies in both sections, considering that it is a general recommendation from APA to articulate introduction and discussion (e.g., it can be inappropriate to introducing new literature in the discussion).

4. The way in which the sample is configured must have bibliographical references to give it validity.

Thank you for your input. We made changes accordingly.

5. The authors must provide information on the representativeness of the sample.

Please see limitations of the current study. We added a new topic on representativeness.

6. The general and specific objectives of the study should be explained with greater precision (only mentioned in line 182,

without specifying).

The information on general and specific objectives is now provided.

7. Table 2 is too broad. The results should be presented more precisely and described in relation to the data in the tables.

The results section was reviewed following your suggestion to increase precision in the description of the main results of the current study.

8. The conclusions are too few. The proposals they mention should be expanded and more detailed with more information.

Thank you for making us aware of this issue. We made some changes in this section.

Round 2

Reviewer 2 Report

Comments and Suggestions for Authors

The authors of the manuscript have made the corrections recommended in the previous round of revisions. Congratulations for their work.